# Unravelling the Importance of Uncertainties in Global-Scale Coastal Flood Risk Assessments under Sea Level Rise

**Jeremy Rohmer [1,\*]**, **Daniel Lincke [2]**, **Jochen Hinkel [2]**, **Gonéri Le Cozannet [1]**, **Erwin Lambert [3]** and **Athanasios T. Vafeidis [4]**

1   BRGM, 45060 Orléans, France; g.lecozannet@brgm.fr
2   Global Climate Forum, 10829 Berlin, Germany; daniel.lincke@globalclimateforum.org (D.L.); hinkel@globalclimateforum.org (J.H.)
3   Royal Dutch Meteorological Institute (KNMI), 3731 De Bilt, The Netherlands; erwin.lambert@knmi.nl
4   Geography Institute, 24098 Kiel, Germany; vafeidis@geographie.uni-kiel.de
\*   Correspondence: j.rohmer@brgm.fr; Tel.: +33-2386-430-92

**Abstract:** Global scale assessments of coastal flood damage and adaptation costs under 21st century sea-level rise are associated with a wide range of uncertainties, including those in future projections of socioeconomic development (shared socioeconomic pathways (*SSP*) scenarios), of greenhouse gas concentrations (*RCP* scenarios), and of sea-level rise at regional scale (*RSLR*), as well as structural uncertainties related to the modelling of extreme sea levels, data on exposed population and assets, and the costs of flood damages, etc. This raises the following questions: which sources of uncertainty need to be considered in such assessments and what is the relative importance of each source of uncertainty in the final results? Using the coastal flood module of the Dynamic Interactive Vulnerability Assessment modelling framework, we extensively explore the impact of scenario, data and model uncertainties in a global manner, i.e., by considering a large number (>2000) of simulation results. The influence of the uncertainties on the two risk metrics of expected annual damage (*EAD*), and adaptation costs (*AC*) related to coastal protection is assessed at global scale by combining variance-based sensitivity indices with a regression-based machine learning technique. On this basis, we show that the research priorities in terms of future data/knowledge acquisition to reduce uncertainty on *EAD* and *AC* differ depending on the considered time horizon. In the short term (before 2040), *EAD* uncertainty could be significantly decreased by 25 and 75% if the uncertainty of the translation of physical damage into costs and of the modelling of extreme sea levels could respectively be reduced. For *AC*, it is *RSLR* that primarily drives short-term uncertainty (with a contribution ~50%). In the longer term (>2050), uncertainty in *EAD* could be largely reduced by 75% if the *SSP* scenario could be unambiguously identified. For *AC*, it is the *RCP* selection that helps reducing uncertainty (up to 90% by the end of the century). Altogether, the uncertainty in future human activities (*SSP* and *RCP*) are the dominant source of the uncertainty in future coastal flood risk.

**Keywords:** future coastal flooding; damage costs; dyke costs; global sensitivity analysis; machine learning

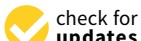



## 1. Introduction

Sea level rise (SLR) is projected to continue and further accelerate over the 21st century [1], which poses a major challenge for coastal areas worldwide in the decades to come [2], particularly regarding the risk of coastal flooding. Future coastal flood damage is expected to increase significantly during the 21st century as sea level rises as shown by several studies both at global [3–6], and at regional scale, e.g., for Europe see [7].

Against this background, estimates of future damage and adaptation costs are key indicators for supporting adaptation planning and policymaking. However, such large-scale damage assessments involve a large number of uncertainties at each step of the

modelling chain from climate drivers to risk metrics, hence resulting in a cascade of uncertainty [8]. In order to support coastal management and planning decisions, it is important that underlying uncertainties are analyzed and documented in order to prevent maladaptation, as outlined by [9].

Despite the importance of understanding uncertainties for decision and policy making, uncertainties in global coastal flood risk assessments have not been explored systematically. The few regional to global-scale assessments that have addressed this question, have relied on a parametric analysis, i.e., by evaluating the relative changes induced by the modification of one *input variable* (i.e., scenario, model parameter, data, etc.) at a time [3,10,11]. Though such approaches can easily and efficiently provide a first order estimate of uncertainty, they only provide narrow insights into the contributions of each uncertain variable, because only a limited number of combinations of input variables are generally accounted for [12].

Here we address this limitation and present the first global scale uncertainty analysis for future coastal flood risk that considers the iteration between uncertain variables. Towards this end, we apply global sensitivity analysis (GSA) [13]. In the context of sensitivity analysis, "global" means that one allows all uncertain variables to vary simultaneously by also taking into account, whenever possible, information on the likelihoods of specific variable values and covariance between variables. We opt for a variance-based GSA, which has shown valuable results when applied to local sites, in France [14], and in the US [15].

The goal thereby is twofold: (i) quantify the impact of the uncertainties; (ii) identify key uncertainties that drive the overall uncertainty in future flood risk. The second objective is of uttermost importance for a full understanding and corroboration of model results [16], because it allows clarifying the role of each uncertainty source, and allows providing a measure of the influence of each uncertainty source (named sensitivity measure) with respect to the variable of interest (e.g., expected annual damage, adaptation costs, etc.).

The GSA is conducted using the coastal flood risk module [3] of the Dynamic Interactive Vulnerability Assessment (DIVA) modelling framework [17]. Among all types of uncertainty, we restrict the analysis to uncertainties that either relate to:

- scenarios (e.g., the choice in the future greenhouse gas emissions using the Representative Concentration Pathways (*RCP*) scenarios 2.6, 4.5 and 8.5, e.g., [18], the future evolution of society that can be represented through the shared socioeconomic pathways (*SSP*) [19]);
- key assumptions, either regarding mathematical relationships (e.g., extreme sea level modelling [20] or vulnerability assessment through depth-damage curves, e.g., [21]), or the degree of variability of some uncertain variable (e.g., sea level projections [22]).

Our application case focuses on the global scale assessments of coastal flood damage and adaptation costs based on the setting of [3]. We extend this previous study by using updated data on mean sea level projection, extreme coastal sea levels [23], and additional uncertainties on datasets (e.g., global population, depth-damage curve) and on modelling approaches of extremes; see further details in Section 2. Among the possible adaptation options (summarized as protect, accommodate, retreat or do nothing by [24]), we focus, following earlier analysis, on the structural measures, which involve protection against flooding through natural (dunes) or artificial (dykes) structures. In particular, economic motivation for raising coastal flood defenses in Europe has extensively been investigated by [25].

The remainder of the paper is organized as follows. Section 2 provides technical details on DIVA, on the statistical method for GSA and on the different modelling uncertainties that are assessed. Section 3 identifies the key drivers of the global-scale assessment by focusing on two risk metrics: the expected annual damage (*EAD*), and the adaptation costs (*AC*) related to dykes. Section 4 discusses the implications of these results as well as the impact of additional residual uncertainties.

## 2. Materials and Methods

### 2.1. DIVA Modelling Framework

The analysis uses the Dynamic Interactive Vulnerability Assessment (DIVA) modelling framework, which has been widely applied to global- and continental-scale assessments of SLR impacts, vulnerability and adaptation. Specifically we use its coastal flood risk and adaptation module, which is described in detail in [3]. Further details on the procedure are also provided in Appendix D. Population and assets exposed to coastal floods are computed using a global coastal segmentation that divides the world's coast into 12,148 variable-length coastal segments based on the Digital Chart of the World [26]. For each coastline segment, a cumulative people exposure function that returns the number of people living below a given elevation level is constructed by superimposing a Digital Elevation Model (DEM) with a spatial population dataset and interpolating piecewise linearly between the given data points. We use Merit DEM with a spatial resolution of 3 s (~90 m at the equator) [27]. For each coastline segment, a cumulative asset exposure function is obtained from the cumulative population exposure function by applying subnational Gross Domestic Product per capita (GDP) rates multiplied by an empirically estimated Assets-to-GDP ratio (*A:GDPr*) [5].

### 2.2. Global Sensitivity Analysis

A global sensitivity analysis (GSA) is performed to quantify the contribution of each uncertainty source to the overall uncertainty of the output variables of the DIVA model. GSA presents the advantages of exploring the sensitivity in a global manner by covering all plausible scenarios for the uncertainties and by fully accounting for possible interactions between them [12,13]. Besides, GSA is model free in the sense that it is applicable to any kind of model (linear, non-linear, and so forth), i.e., without introducing a priori assumptions on its mathematical structure. In the present study, we opt for variance-based GSA, denoted VBSA [13].

Let us denote $f$ as the DIVA model. Consider the $n$-dimensional vector $X$ as a random vector of random input variables $X_i$ ($i = 1, 2, \ldots, n$) that are assigned to the uncertainties described in Table 1. For instance, the input variable for *POP* is assigned a discrete random variable which takes up two values GPW4 or Landscan. Then, the output $Y = f(X)$ is also a random variable (as a function of a random vector). VBSA determines the part of the total unconditional variance Var($Y$) of the output $Y$ resulting from each input random variable $X_i$. Formally, VBSA relies on the first-order Sobol' indices (ranging between 0 and 1), which can be defined as:

$$S_i = \frac{\text{Var}(\text{E}(Y|X_i))}{\text{Var}(Y)} = 1 - \frac{\text{E}(\text{Var}(Y|X_i))}{\text{Var}(Y)}, \tag{1}$$

where E(.) is the expectation operator.

**Table 1.** Uncertain variables and their possible values considered in the global sensitivity analysis.

| Modelling Uncertainty | Values | Number of Values |
|---|---|---|
| Socio-economic development (*SSP*) | SSP1-5 | 5 |
| Greenhouse gas concentrations *(RCP)* | RCP2.6, 4.5, 8.5 | 3 |
| Global population distribution *(POP)* | GPW4 or Landscan | 2 |
| Magnitude of the Regional Sea-Level Rise *(RSLR)* | Median value, 5th and 95th percentile given *RCP* scenario | 3 |
| Logistic depth-damage curves *(DF)* | Half-damage depth 1 or 1.5 m | 2 |
| r-largest annual value *(rGEV)* | Number of r largest values used to fit GEV: 1, 2 or 5 | 3 |
| Subsidence in delta region *(SUBS)* | Included or not | 2 |
| Assets-to-GDP ratio (*A:GDPr*) | Value of 2.8 or 3.8 | 2 |

On the one hand, when the input variables are independent, the index $S_i$ corresponds to the main effect of $X_i$, i.e., the proportion of the variance reduction of $Y$ (i.e., representing the uncertainty in $Y$) that is solely induced by varying $X_i$. The higher the influence of $X_i$, the

lower the variance when fixing $X_i$ (corresponding to the term $\text{Var}(Y|X_i)$ in Equation (1)), hence the closer $S_i$ to one. In general, the sum of all $S_i$ is $\leq 1$ ($i = 1, 2, \ldots, n$); the difference with one is a quantification of the higher order interactions (see [13]).

On the other hand, when dependence exists among the input variables (as it is the case in our study), a more careful interpretation of Equation (1) should be given: in this situation, a part of the sensitivity of all the other input variables correlated with the considered variable contributes to $S_i$, which cannot be interpreted as the proportion of variance reduction due to the only variation in $X_i$, and the sum of all $S_i$ may exceed one. However, interpreting the index $S_i$ as a measure for the reduction in the uncertainty of the output when fixing $X_i$ remains valid in any situation: $S_i$ can be used to identify the input variable that should be fixed (as pointed out by [28]). Whether dependencies are present or not, the index provides a measure of importance (i.e., a sensitivity measure) that is useful for the ranking of the different uncertainties described in Table 1. We adopt this viewpoint in the present study.

In practice, a Monte-Carlo-based approach is used to estimate the main effects (Equation (1)). This is done by randomly sampling multiple configurations of the random variables (typically of several thousands) associated to each uncertainty (defined in Section 2.1). At this stage, dependencies among the uncertainties (in our case corresponding to the *SSP-RCP* combinations, see Section 2.5) can be integrated by using an adequate sampling procedure.

For each of the randomly generated input variables, the time evolution of *AC* and *EAD* are computed. Ideally, this step should be performed by directly using the DIVA model. Yet, due to its computation time cost, this is performed using a machine-learning-based predictive model, which is built to replace the DIVA model, i.e., by using a proxy of DIVA that is of "statistical" nature. To do so, we rely on the random forest (RF) method dedicated to regression [29], which is here particularly suited to the processing of data of mixed types (categorical scenarios, ordinal variables, etc.). The validity of using RF in place of DIVA is assessed via a cross-validation procedure. See further details in Appendix A. The sensitivity indices are then derived by applying a filtering approach as described by [30]: this presents the practical advantage of estimating the main effects even in the presence of input dependencies. Further details are provided in Appendix B.

### 2.3. Uncertain Variables and Values Considered

Table 1 provides an overview of the uncertain variables that we consider in the GSA.

For the Assets-to-GDP ratio (*A:GDPr*), we take the value of 2.8 that has been used in previous assessments of global coastal flood risk [3,5]. This value approximately corresponds to the lower bound of the confidence interval at 90% derived from the regression-based analysis of the 2014 Worldbank data (https://donnees.banquemondiale.org/ accessed on 1 July 2020), namely 2.93. The upper bound of the 90% confidence interval of this analysis (~3.8) is chosen as an alternative value.

To estimate exposure to coastal floods, projected extreme sea levels are obtained by combining projected regional mean sea level (*RSLR*), and present-day extreme sea levels (*ESL*) via the approach described in Appendix C. Projected regional mean sea level is based on the 2019-released SROCC report [23] and depends on the choice in the *RCP* scenario, i.e *RCP*2.6, 4.5 and 8.5. Uncertainty on *RSLR* is related to climate and ice-sheet model uncertainty. In our study, the latter is addressed by using three scenarios: 5th, 50th and 95th percentile derived from the data of [23] by assuming that the underlying probability distribution is Gaussian. *ESL* are derived from the extreme value analysis of the global reanalysis of extreme sea levels calculated using the Global Tide and Surge Model (GTSR) [31]. We focus here on the widely used Generalized Extreme Value (GEV) distribution [32] that can be fitted to the *rGEV* largest values within one year. For *rGEV* = 1, this corresponds to the classical GEV fitted to annual maxima. Two alternatives are considered, namely *rGEV* = 2 or 5 as suggested by [20].

Future exposure is evaluated by applying national population and GDP growth rates of the socioeconomic scenarios to the coastal segments. We use five population and GDP growth scenarios based on the shared socioeconomic pathways (*SSP*s 1–5) [19]. For initialization of the population exposure functions, two datasets of global population distribution are used: GPW4 (https://sedac.ciesin.columbia.edu/data/collection/gpw-v4, accessed on 1 July 2020) or Landscan (https://landscan.ornl.gov/, accessed on 1 July 2020).

Flood extent and water depth are assessed based on spatial analysis, assuming that all areas with an elevation below a certain water level that are hydrologically connected to the sea, are flooded, i.e., following the "bathtub" method [33]. We assume that when the water level is below the protection standard, people and assets are protected, and thus the loss is zero. For people, if there is no protection or the extreme water level is higher than the protection standard, the damage function is identical to the cumulative exposure function.

The translation of the projected water depth into damage is made via a depth-damage curve by using the logistic model used by [3]: this gives the fraction of assets damaged for a given flood depth. The depth-damage curve is parameterized with a parameter specifying the flood depth that destroys 50% of the asset value. Two assumptions for its value are made, namely the value of 1 m used by [3], and 1.5 m defined based on the analysis of two databases [34,35]. See further details in Appendix D.

For coastal segments located in river deltas, additional sea-level change from delta-subsidence is applied. To analyze the uncertainty introduced by this delta subsidence, two values for subsidence rates are used, i.e., of 0 or 2 mm/year (following [3]); note that the maximum value of 2 mm/year corresponds to one standard deviation of all permanent GNSS velocities in the SONEL database after removal of the effects of the global isostatic adjustment using the ICE-5G model [36,37].

Finally, we model adaptation by assuming that the defenses are always provided by protection infrastructure (i.e., dykes, levees or sea-walls). Without adaptation, protection heights are maintained, but not raised, and with adaptation, dykes are raised following a demand function for safety as described by Equation (3) of [3] and applied to each coastal segment.

*2.4. Likelihoods of Variables Values*

Any global uncertainty analysis needs to make assumptions on the likelihood of the values of each uncertain variable considered. While this is straightforward to do for some values, in particular model parameters, this is difficult to do for scenarios, because the primary purpose of scenarios is to describe uncertainties in situations for which possible values and likelihoods are difficult to establish unambiguously, i.e., this is a situation of deep uncertainty [38]. Not explicitly associating likelihoods to scenarios, however, means that these are considered equally likely in a GSA. To deal with this problem, we perform the GSA twice, with different likelihoods associated to input variable values and dependencies between these. Then, we test whether results are robust across both analyses.

The first GSA assumes that scenarios and parameter values are *equally plausible* (i.e., having the same likelihood) except for *RSLR*, because the likelihood of these values is provided by climate model output. Hence, we use likelihood weights of 5, 90 and 5%. This means that, during the random sampling for GSA (see Section 2.2), the scenario for the *RSLR* median value has larger likelihood to occur than the one for the lower (respectively upper) bound. The influence of the assumed likelihood weight values on the GSA results were tested and appeared to be minor (see Supplementary Materials: Section 2.2).

The second GSA assumes that some input variable values are *more likely than others*, as defined in Table 2:

- The r-largest annual value *rGEV* = 5 is chosen as of higher likelihood based on the conclusions of [20];
- A preference is given to the original parametrization of the Logistic depth-damage curves by using a half-damage depth of 1.0 m;

- In light of recent observations by [39], subsidence in delta regions is considered a more plausible scenario;
- Likelihood weights are assigned to combinations of *RCP/SSP*s; this is further detailed below;
- No preference is given, neither for global population distribution, nor for the Assets-to-GDP ratio, in the absence of evidence favoring one scenario over another.

**Table 2.** Definition of most likely scenarios.

| Uncertain Variable | Most Likely Value | Likelihood Weight |
|---|---|---|
| Logistic depth-damage curves | Half-damage depth at 1.0 m | 0.75 |
| Extreme value modelling | $rGEV = 5$ | 0.66 |
| Subsidence in delta region | Included | 0.75 |
| Socio-economic & Greenhouse Gas concentrations | See Table 3 | See Table 3 |

**Table 3.** Definition of the occurrence probability for the Representative Concentration Pathways (*RCP*) scenarios conditional on the shared socioeconomic pathways (*SSP*) scenarios, i.e., each probability value is interpreted as the occurrence of the considered *RCP* scenario given the considered *SSP* scenario. These values should not be interpreted as the probability of *SSP* given *RCP*.

|  | *RCP*2.6 | *RCP*4.5 | *RCP*8.5 |
|---|---|---|---|
| *SSP*1 | 0.18 | 0.34 | 0.48 |
| *SSP*2 | 0.09 | 0.12 | 0.79 |
| *SSP*3 | <0.01 | 0.11 | 0.89 |
| *SSP*4 | 0.12 | 0.18 | 0.70 |
| *SSP*5 | 0.05 | 0.085 | 0.865 |

*2.5. Dependence between Uncertain Variables*

While it is difficult to assign probabilities to *SSP* and *RCP* scenarios, there has been a debate on which *RCP* is more likely given an *SSP* [19]. In the second GSA, we take this into account by assigning occurrence probabilities to *RCPs* conditioned on the occurrence of a given SSP scenario, i.e., P($RCP_{\{2.6,4.5,8.5\}}$ | $SSP_{1-5}$). We estimate P($RCP_{\{2.6,4.5,8.5\}}$ | $SSP_{1-5}$) based on the location of the *RCP-SSP* combination in the diagram "carbon intensity improve rate versus energy intensity improve rate" provided by [19]: the lower both rates, the easier and cheaper it is to transform the economy and hence the higher the conditional probability P($RCP_{\{2.6,4.5,8.5\}}$ | $SSP_{1-5}$) is. More formally, the conditional probability is assumed to be inversely proportional to the product of the carbon intensity and energy intensity improvement rates. Using a normalization so that the probabilities sum to one, this yields Table 3.

Note that this approach of estimating condition probabilities for *RCP* gives non-zero probabilities for some *SSP/RCP* combinations (values in Table 3 are rounded to two decimal places) that are seemingly seen to be infeasible in the literature, most notably for the combination of achieving *RCP*2.6 under *SSP*3. However, the underlying literature that has applied Integrated Assessment Models (IAM) to assess the feasibility of these combinations notes that "the fact that IAMs could not find a solution for some of the 2.6 W/m$^2$ scenarios needs to be distinguished from the notion of infeasibility in the real world." ([19], Page 165). And specifically for the combination of *SSP*3 and *RCP*2.6, they note that "infeasibility, in the case of *SSP*3, is thus rather an indication of increased risk that the required transformative changes may not be attainable due to technical or economic concerns." ([19], Page 165). In any case, assuming a probability of 0 for the *SSP*3/*RCP*2.6 or *SSP*5/*RCP*2.6 combination would not make a big difference as those probabilities are anyway small according to our approach.

## 3. Results

### 3.1. Equally Plausible Values

We analyze the uncertainties in expected annual damage cost (*EAD)* and in the adaptation cost (*AC)* without preference to the likelihood of any input variable.

We first estimate the DIVA model results by accounting for the different combinations as described in Section 2.1. To do so, a total of 2160 DIVA runs were performed. On this basis, we train a RF model at each time step over the time period 2020–2100 to approximate each of the variables of interest. The validation procedure (described in Appendix A) shows that the performance indicator $Q^2$ reaches values from 96.3 to 99.7% (with *mtry* = 8) for *EAD* over time, and >99.9% whatever the time instance for *AC*: this is strong evidence for the high predictive capability of the trained RF models, hence comforting our confidence in the replacement of the DIVA model by the RF models. To compute the sensitivity indices, we randomly generate 50,000 samples of the random variables (assigned to the uncertainties of Table 1) and compute the values of the output variables of interest using the trained RF. This dataset is then used to derive the sensitivity indices as described in Section 2.2. Preliminary convergence tests showed us that 50,000 random samples were sufficient to reach stable sensitivity indices' values (this is also shown by the narrow width of the confidence intervals of the GSA results; see below in Figure 2).

Figure 1b,d shows the mean evolution over time for both output variables of interest (expressed in % of the Gross Domestic Product with reference year of 2014 (https://www. statista.com/statistics/268750/global-gross-domestic-product-gdp/, accessed on 1 July 2020)) together with the uncertainty band whose lower and upper bounds are respectively defined at +/− one standard deviation (red envelope). Both *EAD* and *AC* are projected to increase globally over the 21st century, with the mean *EAD* value reaching 0.7% of the 2014 global GDP and 0.1% for the mean *AC* value. Uncertainty increases over time as well, as indicated by the increase in the width of the uncertainty band, until reaching >+/−80% of the mean value for *EAD*, and ~+/−15% for *AC* by 2100. We note a small decrease of *AC* in 2100 that may be related to the higher variability of inter-annual *AC* fluctuations around its temporal mean as illustrated by five random realizations in Figure 1a (contrary to *EAD*, which presents a smoother time evolution; see Figure 1c): confirming this tendency would however require projections beyond 2100, which is hardly feasible given the current status of data availability.

This significant increase of *EAD* and *AC* and their uncertainties over the 21st century is consistent with previous work [3], though not directly comparable because our study is based on different assumptions for uncertainty representation. It reflects the fact that whatever the future greenhouse gas emissions, sea-level will continue to rise over the 21st century along most of the inhabited shorelines, while exposure is projected to increase globally, although with regional differences [23].

We first analyze the sensitivity for the year 2100 (red barplots in Figure 2). On the one hand, Figure 2a indicates the large impact of *SSP* on the *EAD* uncertainty with an index exceeding 50%: if it were possible to identify unambiguously the future *SSP* scenario, this means that the *EAD* uncertainty (i.e., its variance) would be reduced by >50%. The large impact of *SSP* on the *EAD* uncertainty is expected because within DIVA, *EAD* is computed over a floodplain within which contrasting human development scenarios are projected. The second most important modelling uncertainty appears to be related to the *RCP* scenario with an index >15%. Other uncertainties (*POP*, *rGEV*, *RSLR* and *A:GDPr*) present a sensitivity index of lower magnitude of the order of 5–10%: though low, this order of magnitude remains significant and does not justify to neglect their influence.

On the other hand, Figure 2b shows the large impact of *RCP* on the uncertainty of *AC* with an index exceeding 80%. In this case, *SSP* has a very low sensitivity index hence indicating that being able to properly select a single *SSP* scenario does not influence much the *AC* uncertainty (by 2100). The second major contributor is here *RSLR* with an index of 10%. Other modelling uncertainties appear not to contribute to *AC* uncertainty (to a first level). The large impact of *RCP*s on the uncertainty of *AC* is expected because within DIVA,

locations where adaptation is implemented just raise their defenses in response to sea-level rise, which is highly dependent on *RCP*s by 2100.

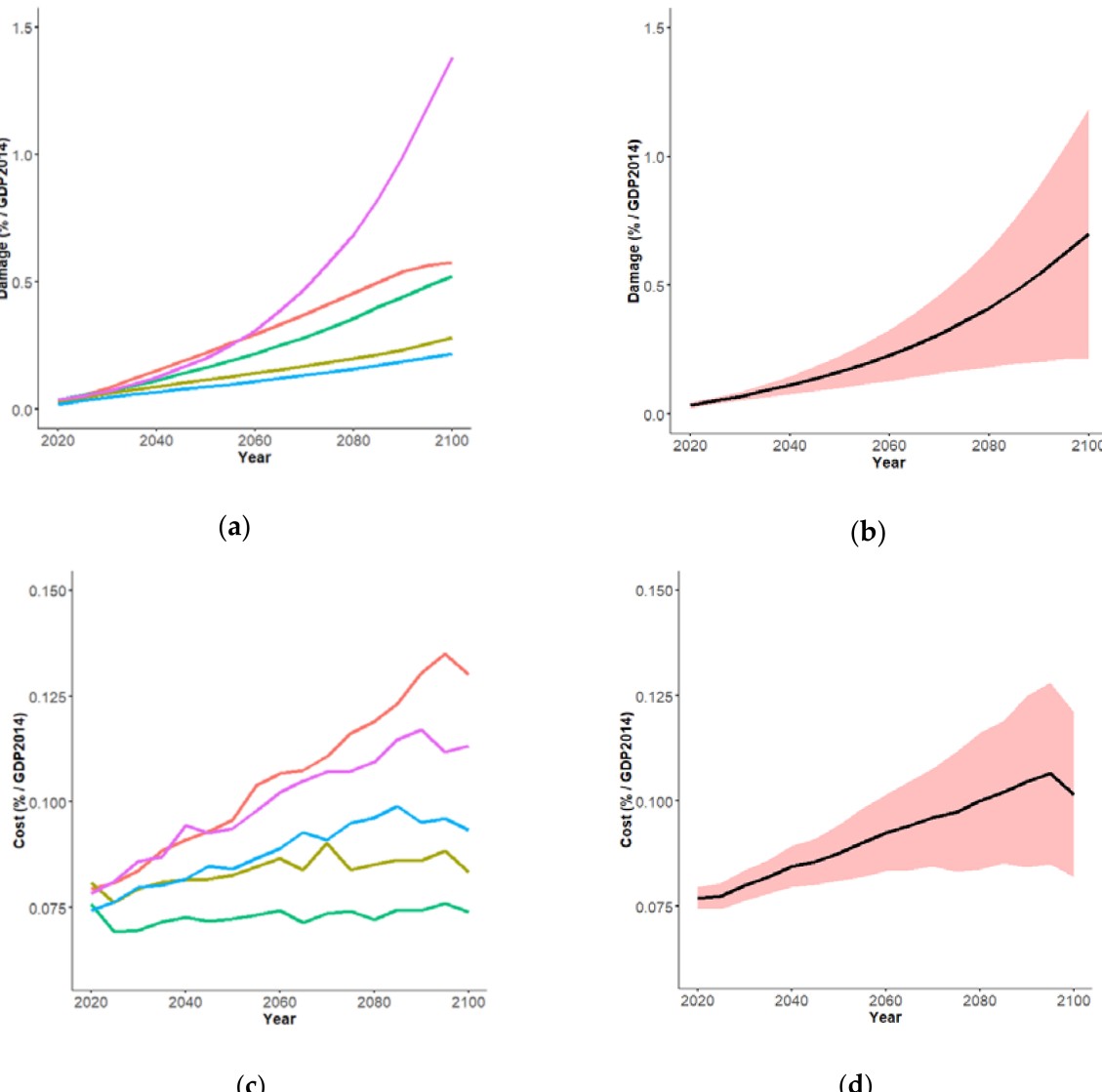

(**a**)  (**b**)

(**c**)  (**d**)

**Figure 1.** Time evolution of the expected annual damage cost *EAD* (Top) and of the adaptation cost *AC* (Bottom) expressed in % of the Gross Domestic Product of 2014 assuming equally plausible modelling scenarios:: (**a,c**) Example of five random realizations; (**b,d**) Mean value (calculated for 50,000 random realizations). The bounds of the uncertainty band are defined at −/+ one standard deviation (red envelope). Note that the vertical y-axis of the *AC* Figures does not start at zero.

The contributions of the uncertainties in the different input variables to the output variable uncertainty vary over time during the 21st century (Figure 3). Note that to ease the interpretation and the comparison with Figure 4, the vertical axis of Figure 3 has been scaled to [0,1], because the sum of all indices reaches ~90% (see Section 2.2). All unscaled results are available in Supplementary Materials: Section 3.

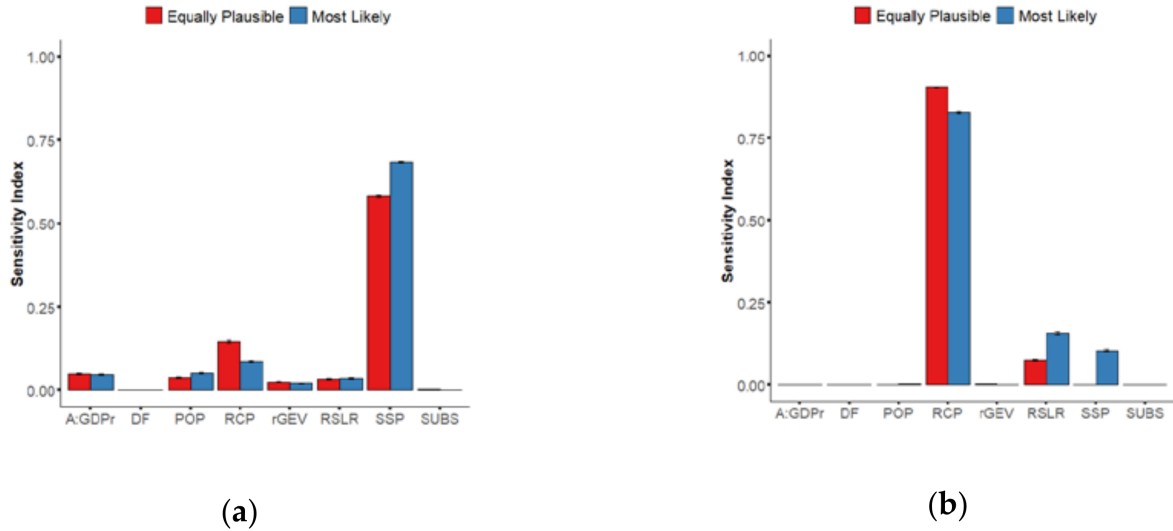

**Figure 2.** Sensitivity indices for *EAD* (**a**) and for *AC* (**b**) by 2100 considering the two assumptions regarding scenarios' likelihood ("equally plausible"-red and with different likelihood weights -blue). The error-bars indicate the 95% confidence interval derived from a bootstrap-based approach (using 250 bootstrap random replicates).

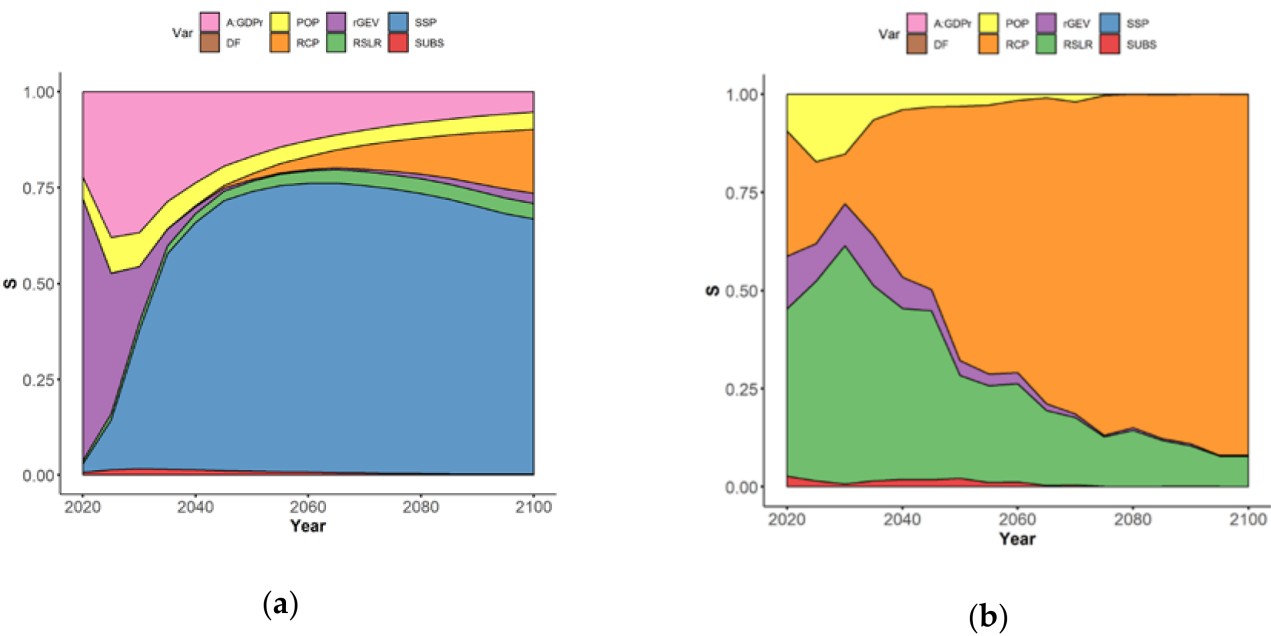

**Figure 3.** Time evolution of the sensitivity indices (denoted S) for *EAD* (**a**) and for *AC* (**b**) considering the situation of equally plausible modelling scenarios. A rescaling has been applied so that the sum of the indices is one.

For *EAD*, Figure 3a shows that:

- In the short term (before 2030–2040, leftmost part of Figure 3a), the uncertainty is mainly controlled by two sources of uncertainty, namely the value of the Assets-to-GDP ratio (rose-colored envelope), the *rGEV* parameter of the extreme value analysis (purple-colored envelope) with sensitivity indices of ~25%, and ~75% respectively;
- After this date, the time evolution of their influence differs: Figure 3a shows the rapidly decreasing *rGEV* influence to low value <1% by 2050, whereas the influence of *A:GDPr* decreases less abruptly, down to values of ~10% by 2050; This decrease is relative to the total uncertainty, which is increasing rapidly after 2040; hence, this decrease simply reflects the growing importance of other sources of uncertainties after 2050;

- The importance of *SSP* scenarios (blue-colored envelope) rapidly increases over time: the sensitivity index reaches large values above 50% after 2030, until driving the whole uncertainty by 2100 with a contribution >50%; again, the slow decrease of the relative contribution of *SSP* to *EAD* uncertainty after 2050 is due to *RCP* becoming a significant source of uncertainty. This large impact of *SSP* is expected because different SSPs have large impacts in the development of the global floodplains;
- The importance of *RCP* scenarios (in orange) is only slowly increasing over time reaching a low index value >5% by 2065–2070; this late emergence of *RCP* scenarios as a source of uncertainty in *EAD* is consistent with the projected times of divergence of sea-level projections: until about 2050, sea-level projections are almost the same whatever the *RCP* scenario due to their high dependence on past greenhouse gas emissions [40];
- The role of the choice in the population database (in yellow) and of *RSLR* (in green) remain moderate with a sensitivity index of 5–8%, and the role of subsidence in delta regions is minor (in red).

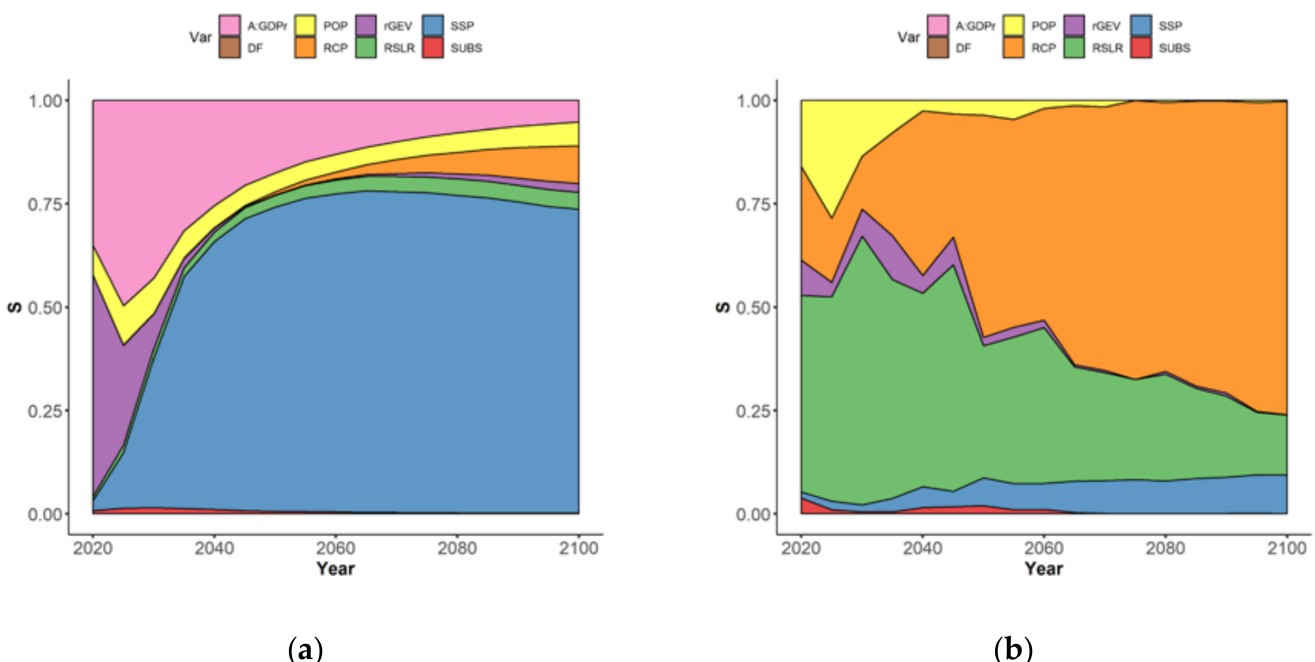

(**a**)                                            (**b**)

**Figure 4.** Time evolution of the sensitivity indices (denoted S) for *EAD* (**a**) and for *AC* (**b**) considering the situation where most likely scenarios have been identified. A small rescaling has been made so that the sum of the indices is one.

For *AC*, the time evolution of the sensitivity indices is completely different from the one of *EAD*:

- In the short term (leftmost part of Figure 3b), the uncertainty related to RSLR (in green) dominates (with an index of >40%);
- The three other most important uncertainties are the population database (in yellow), the extreme modelling parameter (in purple), and the *RCP* scenario (in blue);
- Over time, the importance of *RCP* rapidly exceeds the one of *RSLR*, and reaches >60% by 2050, whereas the one of *RSLR* drops down to <25% after this date;
- The sensitivity index of *POP* and of *rGEV* rapidly decreases down to low value <5% by 2040.

This evolution of *AC* uncertainty reflects the high dependence of adaptation costs to sea-level rise within DIVA. Until 2050, uncertainty in future sea-level rise mostly depends on the differences among climate models (particularly thermal expansion and glaciers

contributions). Beyond 2050, *RCP* emerges as a significant uncertainty underpinning future sea-levels.

### 3.2. With Most Likely Values

The same procedure is now conducted by considering a situation where, for each uncertainty, a scenario has been assigned a higher likelihood than others (see Tables 2 and 3). Figure 2a (blue barplots) shows that, accounting for the scenario occurrence likelihood only moderately modifies the sensitivity indices except for *SSP* (respectively *RCP*) with an increase (respectively decrease) of ~10% for *EAD*. These changes remain of moderate order of magnitude, which shows that the tendencies in the sensitivity evolution (highlighted in Section 3.1) remain robust even when differentiating likelihoods. The impact for *AC* is more noticeable: *SSP* influence reaches (approximately) the one of *RSLR* by 2100.

Figure 4 shows that the time evolution of the respective modelling uncertainties follows the same tendency as for Figure 3 (i.e., the shape of the trend over time remains similar). Note that to ease the interpretation and the comparison with Figure 3, the vertical axis of Figure 4 has been scaled to [0, 1], because the sum of all indices reaches ~110–115% (see Section 2.2). All unscaled results are provided in Supplementary Materials: Section 3.

Accounting for higher likelihood appears to strengthen the role of *SSP* for *EAD* (with an increase of the sensitivity index by almost 10% over time after 2040), as well as for *AC* (with an emergence of a significant contribution after 2040–2045; with an index value >5%). Conversely, a decrease of the *RCP* role is outlined in both cases by an order of magnitude equivalent to the increase in the *SSP* contribution, which is a direct consequence of considering dependence in Table 3.

The influence of the assumed likelihood weight values were tested by setting the conditional probability values (of Table 3) below 10% to zero. On the one hand, the RCP2.6-SSP2, RCP2.6-SSP3, RCP2.6-SSP5, and RCP4.5-SSP5 combinations became unlikely and were not accounted for in the sampling procedure. On the other hand, the likelihood weight values of RCP4.5-SSP2, RCP85-SSP2, and RCP8.5-SSP5 increased (up to 13.5%). Compared to Figure 4, these changes only lead to minor differences (see Supplementary Materials: Section 2.1) in the value of the sensitivity measures (no more than 5%). The time evolution of the sensitivity measure appears to be the same as well as the importance ranking: our conclusions can thus be considered little affected by assumed changes in likelihood weights and by the removal of the afore-mentioned *RCP-SSP* combinations.

Finally, we also note the enhanced role of *RSLR*, with quasi- constant contribution of ~10% (after 2060), compared to the situation of equally plausible scenarios.

## 4. Discussion and Conclusion

### 4.1. Summary and Implications

The afore-described VBSA results should be translated into research priorities in terms of future data/knowledge acquisition to reduce the uncertainty in damage and adaptation costs. These priorities differ depending on the considered time horizon.

The results suggest that, in the short term (<2040), the uncertainty in *EAD* could be reduced by acting on two key modelling uncertainties, namely the parametrization of the extreme value distribution (the *rGEV* largest annual values used to fit the GEV distribution), and the ratio to translate damage into costs (the Assets-to-GDP ratio). In the long term, the uncertainty is expected to be largely reduced if the scenario *SSP* could be unambiguously identified even when accounting for the occurrence likelihood of *SSP-RCP* combinations (as confirmed by the experiment in Section 3.2).

For *AC*, improving knowledge of *RSLR* appears to be the most critical in the short- and mid-term (before 2050). After this date, *RCP* (and *SSP* if combination likelihoods are accounted for) becomes the major driver of uncertainty (Figure 4b): this means that mitigation of climate change helps here to reduce the uncertainty in adaptation costs.

Finally, it should be noted that the transition from short to long term does not necessarily follow a linear trend, i.e., the importance of the modelling uncertainties evolve in a non-linear fashion over time, with a regime shift at 2030/2040.

### 4.2. Residual Uncertainties

The present study covers the main modelling uncertainties of DIVA. However, some residual uncertainties remain. Their potential influence is discussed here as well as some suggestions for improvement.

First, the choice of the Digital Elevation Model was not accounted for though it is known to influence the assessment results as discussed by [3] at a global scale, and still shown recently by [41]. Due to the overwhelming importance of this uncertainty, the results of GSA would have been little informative and we chose to focus on the Merit DEM because: (i) it has a satisfactory spatial resolution (of 3 s, i.e., ~90 m at the equator); (ii) it has a complete global coverage; (iii) it corrects some of the known systematic errors from the alternative widely used SRTM [42].

Second, the vertical ground motion was only accounted for in regions that are particularly subject to rapid subsidence, i.e., in delta plains. Our results suggest that this process hardly affects the uncertainty (of low magnitude of 1–2% for both *AC* and *EAD*) compared to the other sources of uncertainty. Yet, this result may be related to the relatively moderate range of the subsidence rate (assumed here between 0 and 2 mm/year). Other regions can be affected by larger subsidence (or uplift), and higher values than 2 mm/year can be found locally (e.g., in Manilla [43]; in Djakarta [44]), but they strongly depend on ongoing and future groundwater extraction, so that extrapolating them in the future can arguably be defended (as shown for instance in Shanghai by [45]). As a future direction of the present study, extending the analysis to other regions would be of interest provided that two difficulties are addressed: (i) setting up a global-scale database for present-day subsidence rates. The Copernicus Ground motion service or the network of Permanent Service for Mean Sea Level (https://www.psmsl.org/, accessed on 13 January 2021) could here play a major role; (ii) defining plausible scenarios of the future evolution of subsidence rates. The latter is strongly dependent on the availability of geological data and of urbanization projections that require appropriate modelling developments (see e.g., [46] for 10 Mediterranean countries).

Third, we focused on one type of flooding algorithms, i.e., the widely-used bathtub approach [33], which has been reported to overestimate impacts in some locations. Alternative approaches could be envisaged like the quasi-dynamic approach of [47], or the modelling of inland water attenuation (see the approach by [48]), though this type of uncertainty is expected to be lower, at least at large/global scale, compared to the others as highlighted by [10] in their regional-scale study.

Fourth, the depth-damage relation was modelled by a logistic model (see Appendix D). We tested the influence of the half-damage depth (by assuming two scenarios: 1 or 1.5 m) Our GSA results indicate that the importance of this uncertainty is lower compared to the others. To some extent, this result is consistent with the results of the parametric study performed by [10] on their regional-scale site. Yet, our result does not include the influence of the choice in the type of depth-damage model: the influence of this modelling uncertainty is expected to be non-negligible given the large spectrum of different model types (see e.g., [49]) and should be included in future analyses.

Finally, our results show the non-negligible role of how extremes are modelled in the short term. This goes in the same line than the findings of [20] and of [14] at a more local scale. Yet, our study only focuses on one aspect of this problem (the number of r largest value used to fit GEV) and the analysis should be extended by including the fitting error of such method, as well as additional processes like the wave contribution to extremes (as shown on a local/regional scale by [10], and on a quasi-global scale by [50]), and the tide-surge interaction [51] as well as the influence of SLR on tides [52]. Besides, extreme

value analysis is restricted in our study to one single dataset of coastal water level hindcasts (i.e., the one of [31]), and its uncertainties are not quantified here.

A number of other sources of uncertainty can be mentioned, such as the accuracy of the tidal levels with respect to the DEM reference, the different factors affecting the relative sea level variability (in particular the changes in ocean circulation and density; see e.g., [53]), the actual protection levels granted by current coastal protections, the effects of changing shorelines on flood hazards, as they may create new pathways during storms, etc. These uncertainties have not been considered here, but the developed procedure is flexible enough to incorporate these additional elements in future works.

**Supplementary Materials:** The following are available online at https://www.mdpi.com/2073-444 1/13/6/774/s1. The description of the data and codes (and the procedure to download them) are provided in Supplementary Materials: Section 1. Additional sensitivity tests on the influence of the likelihood weights are provided in Supplementary Materials: Section 2. Unscaled GSA results are provided in: Supplementary Materials: Section 3.

**Author Contributions:** Conceptualization, J.R.; methodology, J.R., D.L., J.H., G.L.C.; software, J.R., D.L.; formal analysis, J.R.; investigation, J.R., D.L.; resources, D.L., E.L., A.T.V.; data curation, J.R., D.L., E.L., A.T.V.; writing—original draft preparation, J.R.; writing—review and editing, all; project administration, G.L.C.; funding acquisition, G.L.C., J.H. All authors have read and agreed to the published version of the manuscript.

**Funding:** All authors have received funding from the INSeaPTION project which is part of ERA4CS, an ERA-NET initiated by JPI Climate (Grant 690462).

**Institutional Review Board Statement:** Not Applicable.

**Informed Consent Statement:** Not Applicable.

**Data Availability Statement:** The data presented in this study are openly available in https://github.com/daniellincke/DIVA_paper_uncertainty 13 January 2021. See also further details in Supplementary Materials: Section 1.

**Acknowledgments:** We thank the participants to the WRCP Climate Services workshop (11–13 November 2019, Orleans, France) and to the second CoastMIP/ISIpedia workshop on large scale coastal flood risk assessment under sea-level rise (10–11 June 2020, virtual) to fruitful discussions

**Conflicts of Interest:** The authors declare no conflict of interest.

## Appendix A. Random Forest Regression

Random Forest (RF) is a non-parametric regression technique based on a combination (ensemble) of tree predictors (using regression tree as formally introduced by [54]). Each tree in the ensemble (forest) is built based on the principle of recursive partitioning, where the parameter space is recursively partitioned into a set of rectangular areas. The partition is created such that observations with similar response values are grouped.

The splitting threshold value $z$ is selected in accordance with squared residuals minimization algorithms [0]. Let us denote $Y_i$ the result of the DIVA simulation given the $i$th vector of $p$ input variables: $X_i = \left( X_i^{(1)}, \ldots, X_i^{(p)} \right)$ with $i = 1, \ldots, n$. A cut of a given area $A$ is defined by the pair $(j, z)$ where $j = (1, \ldots, p)$ and $z$ is the position along the $j$th coordinate given the limits of $A$. Let us define $C_A$ the set of all possible cuts for $A$. The partition aims at maximizing the following criterion over $C_A$:

$$L(j,z) = \frac{1}{N(A)} \sum_{i=1}^{n} \left( Y_i - \overline{Y_A} \right)^2 I_{(X_i \in A)} - \frac{1}{N(A)} \sum_{i=1}^{n} \left( Y_i - \overline{Y}_{A_L} I_{\left( X_i^{(j)} < z \right)} - \overline{Y}_{A_R} I_{\left( X_i^{(j)} \geq z \right)} \right)^2 I_{X_i \in A}, \tag{A1}$$

where $N(A)$ is number of elements in the given area $A$, $A_L = \left\{ X \in A : X^{(j)} < z \right\}$ and $A_R = \left\{ X \in A : X^{(j)} \geq z \right\}$ are the new areas after splitting, $\overline{Y_A}$ (respectively $\overline{Y}_{A_L}$, and $\overline{Y}_{A_R}$) is the average of the results $Y_i$ that belong to $A$ (respectively $A_L$ and $A_R$), and $I_{(B)}$ is the indicator function that reaches 1 if $B$ is true and 0 otherwise.

After the partition is completed (until subdivision no longer decreases the splitting criterion, or until a minimum node size $n_{\text{size}}$ is reached), a constant value of the response variable is predicted within each area (mean value for regression).

RF builds on the same principles as decision tree models but extends them by adding a random character to the construction process at two levels: 1. each tree is constructed using a different bootstrap sample of the data; 2. each node is split using the best among a subset of *mtry* randomly selected predictor variables. This random procedure is repeated $n_{\text{tree}}$ times, i.e., the RF model is set up using an ensemble of $n_{\text{tree}}$ different tree models.

The validity of using RF model to model the DIVA results is tested by evaluating the predictive capability of the trained RF, i.e., the capability of the RF model to correctly predict the DIVA output given "unseen" input configurations of the input variables. This validation exercise is performed based on 10-fold cross-validation procedure [55] by computing the $Q^2$ coefficient defined as follows:

$$Q^2 = 1 - \sum_{i=1}^{n} \frac{(Y_i - \hat{Y}_i)^2}{(Y_i - \overline{Y})^{2\prime}} \tag{A2}$$

where $\hat{Y}_i$ is the RF-based prediction of $Y_i$, and $\overline{Y}$ is the mean value of $Y$.

The closer $Q^2$ to one, the more satisfactory the validation, the higher the predictive capability of the RF model. To further increase the predictive capability, we tune the value of *mtry* to reach the maximum $Q^2$ value. Figure A1 depicts the time evolution of $Q^2$ as a function of *mtry*. This shows that *mtry* = 8 leads to a very satisfactory predictive capability of the RF model for both *EAD* and *AC*: $Q^2$ ranges from 96.3 to 99.7% for *EAD* over time, and >99.9% whatever the time instance for *AC*. The influence of the other RF parameters (i.e., $n_{\text{tree}}$ and $n_{\text{size}}$) on the predictive capability was also tested and appeared to be minor compared to the one of *mtry*: they were fixed to constant values $n_{\text{tree}}$ = 1000 and $n_{\text{size}}$ = 5.

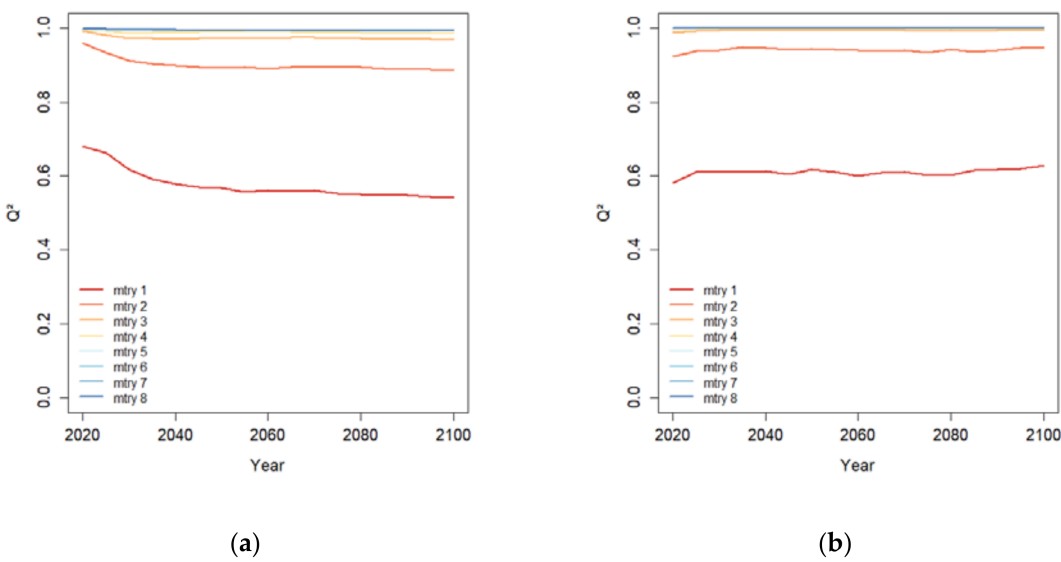

(a)  (b)

**Figure A1.** Time evolution of the performance indicator $Q^2$ estimated using a 10-fold cross-validation procedure for *EAD* (**a**) and for *AC* (**b**) considering different *mtry* values.

## Appendix B. Filtering Algorithm for Estimating the Sensitivity Indices

We rely on the filtering procedure of [30] for computing the first-order Sobol' indices as described in Equation (1). The procedure exploits the equivalence $\mathrm{Var}(\mathrm{E}(Y|X_i)) = 1 - \mathrm{E}(\mathrm{Var}(Y|X_i))$ by following the steps:

1.  Random sampling of the uncertainty input space. No particular algorithm is here required, and dependencies can here be taken into account using appropriate random sampling scheme;

2.  Partition of the input parameter space into clusters. This can be done in different manners, for instance using an equi-probable partitioning [30]. Alternatives can focus on clustering techniques; for instance K-means algorithm with a fixed number of samples, or a simple systematic regular partition. In the present study, the partition corresponds, by construction, to the selection of a given modelling scenario;

3.  Computation of the local conditional variance, i.e., $\text{Var}(Y|X_i)$ for each cluster. This provides a quantification of the evolution of the local variance in the domain of variation of the input parameters;

4.  Estimation of the local Sobol' indices, i.e., $1 - \text{Var}(Y|X_i)/\text{Var}(Y)$. Depending on the value at which the parameter is fixed, the local variance reduction can be high or low;

5.  The Sobol' indices are then obtained by computation of the expectation values of the local Sobol' Indices as described by Equation (1).

**Appendix C. Projections of Extreme Sea Levels**

We use the global reanalysis of sea levels calculated using the Global Tide and Surge Model (GTSR) [31]. The hourly time series are pre-processed by subtracting the annual maxima of sea levels at each location. These data were used to fit a Generalized Extreme Value (GEV) distribution [32]. From this analysis, the $m$-year return level for present day, denoted $rl_m$, can be calculated as follows:

$$rl_m = \mu - \frac{\sigma}{\xi}\left(1 - \left(-\log\left(1 - \frac{1}{m}\right)\right)^{-\xi}\right) \tag{A3}$$

where $\mu$, $\sigma$, and $\xi$ are the GEV location, scale and shape parameters that are estimated through maximum likelihood estimation [32].

The projection of the $m$-year return level is assessed at a future time instant $t$ by shifting the location parameter $\mu$ in Equation (A3) by an amount corresponding to the magnitude of regional sea level rise $RSLR(t, q, RCP)$: it corresponds to the quantile at level $q$ of the Gaussian law fitted to the data of [23] conditioned on the choice in the $RCP$ scenario. By assuming that only the location parameter is influenced by $RSLR$, we implicitly assume that $RSLR$ has no impact on future storm characteristics or tidal elevations.

These results are projected on the DIVA coastal segments by applying a nearest neighbor search algorithm, and are used as inputs to compute the coastal flood risk (see Appendix D).

The afore-described procedure is extended by analyzing the $rGEV$ largest values within one year (instead of the annual maxima, which correspond to $rGEV = 1$) by following the approach described by [20] (and references therein).

**Appendix D. Coastal Flood Risk in DIVA**

The following Appendix is based on the description of DIVA flooding module provided by [3,17]. See also Supplementary Materials for data and codes availability.

Flood risks are assessed using the DIVA model with a refined flooding algorithm (Version 2.4.1). People and assets exposed to coastal flood events are computed using a global coastal segmentation that divides the world's coast into 12,148 variable-length coastal segments based on the Digital Chart of the World [26].

For each coastline segment, a cumulative population exposure function (denoted $e_p$) that gives the number of people living below a given elevation level $x$ is constructed by superimposing a DEM with a spatial population dataset and interpolating piecewise linearly between the given data points. Only grid cells that are hydrologically connected to the coast are considered. From areas below 1 m of elevation, the areas covered by coastal wetlands are subtracted, because these are uninhabitable. For each segment, a cumulative

asset exposure function (denoted $e_a$) is obtained by applying subnational per capita GDP rates to the population data multiplied by an empirically estimated Assets-to-GDP ratio (*A:GDPr*) [5]. Future exposure is calculated by applying national population and GDP growth rates of the shared socioeconomic paythway scenarios (*SSP*) to the coastal segments.

The damage function for population equals the cumulative exposure function, i.e., $d_p(x) = e_p(x)$. For assets, the damage is assumed to be related to the water depth of flooding. A logistic depth-damage function is used to compute the fraction of assets, denoted vul(*h*), that are damaged for a given flood depth *h*:

$$\text{vul}(h) = h/(h+a), \tag{A4}$$

where *a* is the half-damage depth specifying the flood depth that destroys 50% of the asset value.

The damage to assets induced by a flood of height *x* is then computed by integrating from elevation level 0 to *x* over the product of the depth-damage function applied to the water depth $x-y$ and the derivative of the cumulative exposure function $e'_a$ applied to the elevation level *y* as follows:

$$d_a(x) = \int_0^x \text{vul}(x-y)e'_a(y)dy, \tag{A5}$$

With dykes, we assume that the damage is 0 for floods with a height below the top of the dyke. Finally, the population flooded and the flood damages are computed as a mathematical expectation of the population and assets damage functions as follows:

$$\int_{x_{dyke}}^{x_{max}} d(x)f(x)dx, \tag{A6}$$

where $x_{dyke}$ is the dyke height, and $x_{max}$ is the maximum extreme sea level to be taken into account (here assumed to be the 10,000-year return level). The probability density function $f(.)$ is the one of extreme sea levels and is derived from the analysis described in Appendix C.

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
