# Peer review of "Unravelling the Importance of Uncertainties in Global-Scale Coastal Flood Risk Assessments under Sea Level Rise"

_water, doi:10.3390/w13060774_

Round 1

Reviewer 1 Report

The paper reports a study dealing with the evaluation of the uncertainties in coastal flood risk assessment at a global-scale. The DIVA framework is used to model the effects of Sea Level Rise on impact, vulnerability and adaptation measures. Then a Global Sensitivity Analysis is used to determine the sensitivity of the outputs of the DIVA model with respect to the uncertainties in source variables.

The study is very interesting since it quantifies the uncertainties in risk assessment modelling chain, which are often neglected in such kind of research. The manuscript is well structured and written. The introduction provides all the relevant references and information useful to understand the research and the aim of the study. Methods and results are clearly presented and discussed. The paper is suitable for publication, despite some very minor revisions could be addressed, as follows:

  • please, note that some references have been written twice (e.g. L. 109, 125...).
  • table 1 the symbol could be between brackets near the variable, in order to make the table more readable
  • Figure 2: the legend is too small

Author Response

We would like to thank Reviewer #1 for the constructive and highly valuable comments. We agree with most of the suggestions and, therefore, we have modified the manuscript to take on board their comments. In the attached document, we recall the reviews and we reply to each of the comments in turn (outlined in blue).

Reviewer 2 Report

It is well-written rhetorically but the scientific readability is low. A fatal problem is that it discussed uncertainties without disclosing important engineering details used in analysis. For example, in line 196, it stated "...water level is below the protection standard, people and assets are protected, and thus the loss is zero". However, how flood extent and water depth were obtained and how the protection standard was chosen were not clearly explained. Besides, it is more mathematical than explanatory. For example, the discussion and conclusion did not provide any clear information about how the depth-damage curve used in this study affects uncertainties? Furthermore, the following question deserves an answer. Sea level rise over the coming centuries may affect tidal characteristics substantially and could even affect storm surge. How these possibilities were accounted for in the uncertainty analysis?

Author Response

We would like to thank Reviewer #2 for the constructive and highly valuable comments. We agree with most of the suggestions and, therefore, we have modified the manuscript to take on board their comments. In the attached document, we recall the reviews and we reply to each of the comments in turn (outlined in blue).

Reviewer 3 Report

This topic is worthy of research.
However, after reading most of the references that appear in the manuscript, and other publications on the same topic, I cannot find in this manuscript much content of new and useful scientific value for the international scientific community.

In fact, it is a review article largely based on self-references (particularly two of them), with updated data on mean sea level projection, extreme coastal sea levels, and additional uncertainties on datasets. The methodology and assumptions are essentially derived from self-references, basically without adding innovation with useful scientific relevance.

In qualitative terms, the assumptions made and the results presented in Figures 3 and 4 can be accepted, as well as other possible ones. Anyway, in quantitative terms, with the information and general descriptions provided throughout the text, the results can hardly be gauged by interested readers (the basic principle of scientific literature).

Note that the various assumptions made, with projections and simulations not sufficiently reliable, necessarily lead to results of great uncertainty and of little use.

Also a note for some gaps throughout the text, of which several citations are examples.

Still a final note for the citations; a scientific article cannot be confused with a repository of self-references (at least 26), of which 16 are from the same author.

Therefore, this manuscript can hardly be framed within the scope of a scientific article. At most, it can be accepted as a Review or Technical Note, after a careful revision, including essential data provision (possibly as supplementary material), a sufficiently detailed description of the methodology used to obtain the results, and a great reduction of self-citations.

Author Response

We would like to thank Reviewer #4 for the constructive and highly valuable comments. We have modified the manuscript to take on board their comments. In the attached document, we recall the reviews and we reply to each of the comments in turn (outlined in blue).

Reviewer 4 Report

The authors assessed the uncertainties of coastal flood damage and adaptation costs that are related to sea-level rise over a globe scale under 21st century. It is a nice study, which help deepened our understanding of the uncertainties of sea level rise and associated coastal flood damage and adaptation costs. Hence I recommend publication of this manuscript pending on minor revision. 

Besides the uncertainty sources indicated by the authors, ocean circulation changes could also play a role. Chen et al. (2019) suggested that uncertainties in the change of the Atlantic meridional overturning circulation (AMOC) can cause uncertainties in regional sea level change over a global scale. While climate models may even overlook the possibility of a collapse of future AMOC (Liu et al. 2017). I would like to suggest the authors adding discussions regarding the change of ocean circulation in contributing to the projection of sea level change.

Chen, C., Liu, W., and Wang, G. (2019). Understanding the uncertainty in the 21st century dynamic sea level projections: The role of the AMOC. Geophysical Research Letters, 46, 210– 217.

Liu, W., Xie, S.P., Liu, Z. and Zhu, J. (2017). Overlooked possibility of a collapsed Atlantic Meridional Overturning Circulation in warming climate. Science Advances3, e1601666.

Author Response

We would like to thank Reviewer #4 for the constructive and highly valuable comments. We agree with most of the suggestions and, therefore, we have modified the manuscript to take on board their comments. In the attached document, we recall the reviews and we reply to each of the comments in turn (outlined in blue).

Round 2

Reviewer 2 Report

Improved significantly

Author Response

We would like to thank Reviewer 2 for his/her positive comment. Minor spell was also checked.

Reviewer 3 Report

I agree with the authors regarding the amount of data used and the scale of analysis.
What should not be confused is an extension and scale of analysis with a scientific novelty in conceptual and methodological terms (which is what really matters).

Indeed, in conceptual and methodological terms, this manuscript falls within the scope of other publications, many of which are from the same authors (17/56).

It is true that in the review process more information on the tools used is provided, mainly in an appendix, and this is positive. However, with the information made available, the results are unlikely to be reproduced by an interested reader (basic principle of scientific literature), which makes them of little use.

Taking into account the various assumptions made and the projections and simulations carried out, can the authors state that the results are sufficiently substantiated and reliable?

In short, I still believe that this manuscript is far from representing cutting-edge work in scientific terms. However, given the revision produced, including the new elements made available, and taking into account the significant number of consulted publications and the scope and extension of the analyses carried out, I believe that the current content is sufficient for a possible publication as a Review.

Author Response

We would like to thank Reviewer 3 for taking time for reviewing our work. We have modified the manuscript to take on board some of the comments. In the attached document, we recall the reviews and we reply to each of the comments in turn.
